# ReD-GCN: Revisit the Depth of Graph Convolutional Network

## Abstract

Finding the proper depth $d$ of a GNN that provides strong representation power has drawn significant attention, yet nonetheless largely remains an open problem for the graph learning community. Although noteworthy progress has been made, the depth or the number of layers of a corresponding GCN is realized by a series of graph convolution operations, which naturally makes $d$ a positive integer ($d \in \mathbb{N}+$). An interesting question is whether breaking the constraint of $\mathbb{N}+$ by making $d$ a real number ($d \in \mathbb{R}$) can bring new insights into graph learning mechanisms. In this work, by redefining GCN's depth $d$ as a trainable parameter continuously adjustable within $(-\infty, +\infty)$, we open a new door of controlling its expressiveness on graph signal processing to model graph homophily/heterophily (nodes with similar/dissimilar labels/attributes tend to inter-connect). A simple and powerful GCN model ReD-GCN, is proposed to retain the simplicity of GCN and meanwhile automatically search for the optimal $d$ without the prior knowledge regarding whether the input graph is homophilic or heterophilic. Negative-valued $d$ intrinsically enables high-pass frequency filtering functionality for graph heterophily. Variants extending the model flexibility/scalability are also developed. The theoretical feasibility of having a real-valued depth with explainable physical meanings is ensured via eigen-decomposition of the graph Laplacian and a properly designed transformation function from the perspective of functional calculus. Extensive experiments demonstrate the superiority of ReD-GCN on node classification tasks for a variety of graphs. Furthermore, by introducing the concept of eigengraph, a novel graph augmentation method is obtained: the optimal $d$ effectively generates a new topology through a properly weighted combination of eigengraphs, which dramatically boosts the performance even for a vanilla GCN.

## 1 Introduction

Graph convolutional network (GCN) (Kipf & Welling, 2016; Veličković et al., 2017; Hamilton et al., 2017) has exhibited great power in a variety of graph learning tasks, such as node classification (Kipf & Welling, 2016; Luan et al., 2019; 2022a), link prediction (Zhang & Chen, 2018), community detection (Chen et al., 2020), and many more. Since the representation power of GCN is largely determined by its depth, i.e., the number of graph convolution layers, tremendous research efforts have been made on finding the optimal depth that strengthens the model's ability for downstream tasks. Upon increasing the depth, the over-smoothing issue arises: a GCN's performance is deteriorated if its depth exceeds a uncertain threshold (Kipf & Welling, 2016). It is unveiled in (Li et al., 2018) that a graph convolution operation is a special form of Laplacian smoothing (Taubin, 1995). Thus, the similarity between the graph node embeddings grows with the depth so that these embeddings eventually become indistinguishable. Various techniques are developed to alleviate this issue, e.g., applying pairwise normalization can make distant nodes dissimilar (Zhao & Akoglu, 2019), and dropping sampled edges during training slows down the growth of embedding smoothness as depth increases (Rong et al., 2019).

Other than the over-smoothing issue due to large GCN depth, another fundamental phenomenon widely existing in real-world graphs is homophily and heterophily. In a homophilic graph, nodes with similar labels or attributes tend to inter-connect, while in a heterophily graph, connected nodes usually have distinct labels or dissimilar attributes. Most graph neural networks (GNNs) are developed based on homophilic assumption (Yang et al., 2016), while models able to perform well on heterophilic

graphs often need special treatment and complex designs (Bianchi et al., 2021; Zhu et al., 2020). Despite the achievements made by these methodologies, little correlation has been found between the adopted GNN model's depth and its capability of characterizing graph heterophily.

For most GNNs, if not all, the depth needs to be manually set as a hyper-parameter before training, and finding the proper depth usually requires a considerable amount of trials or good prior knowledge of the graph dataset. Since the depth represents the number of graph convolution operations and naturally takes only positive integer values, little attention has been paid to the question whether a non-integer depth is realizable, and if yes, whether it is practically meaningful, and whether it can bring unique advantages to current graph learning mechanisms.

This work revisits the GCN depth from spectral and spatial perspectives and explains the interdependencies between the following key ingredients in graph learning: the depth of a GCN, the spectrum of the graph signal, and the homophily/heterophily of the underlying graph. Firstly, through eigen-decomposition of the symmetrically normalized graph Laplacian, we present the correlation between graph homophily/heterophily and the eigenvector frequencies. Secondly, by introducing the concept of eigengraph, we show the graph topology is equivalent to a weighted linear combination of eigengraphs, and the weight values determine the GCN's capability of capturing homophilic/heterophilic graph signals. Thirdly, we reveal that the eigengraph weights can be controlled by GCN's depth, so that an automatically tunable depth parameter is needed to adjust the eigengraph weights into the designated distribution in match of the underlying graph homophily/heterophily.

To realize the adaptive GCN depth, we extend its definition from a positive integer to an arbitrary real number with theoretical feasibility guarantees from functional calculus (Shah & Okutmuştur, 2020). With a trainable depth parameter, we propose a simple and powerful model, Redefined Depth-GCN (ReD-GCN), with two variants. Extensive experiments demonstrate the automatically optimal depth searching ability, and it is found that negative-valued depth plays the key role in handling heterophilic graphs. Systematical investigation on the optimal depth is conducted in both spectral and spatial domains. It in turn inspires the development of a novel graph augmentation methodology. With clear geometric explanability, the augmented graph structure possesses supreme advantages over the raw input topology, especially for graphs with heterophily. The main contributions of this paper are summarized as following:

- The interdependence between negative GCN depth and graph heterophily is discovered; In-depth geometric and spectral explanations are presented.

- A novel problem of automatic GCN depth tuning for graph homophily/heterophily detection is formulated. To our best knowledge, this work presents the first trial to make GCN's depth trainable by redefining it on the real number domain.

- A simple and powerful model RED-GCN with two variants (RED-GCN-S and RED-GCN-D) is proposed. A novel graph augmentation method is discussed.

- Our model achieves superior performance on semi-supervised node classification tasks on 11 graph datasets.

## 2 PRELIMINARIES

**Notations.** We utilize bold uppercase letters for matrices (e.g., $\mathbf{A}$), bold lowercase letters for column vectors (e.g., $\mathbf{u}$) and lowercase letters for scalars (e.g., $\alpha$). We use the superscript $\top$ for transpose of matrices and vectors (e.g., $\mathbf{A}^\top$ and $\mathbf{u}^\top$). An attributed undirected graph $\mathcal{G} = \{\mathbf{A}, \mathbf{X}\}$ contains an adjacency matrix $\mathbf{A} \in \mathbb{R}^{n \times n}$ and an attribute matrix $\mathbf{X} \in \mathbb{R}^{n \times q}$ with the number of nodes $n$ and the dimension of node attributes $q$. $\mathbf{D}$ denotes the diagonal degree matrix of $\mathbf{A}$. The adjacency matrix with self-loops is given by $\tilde{\mathbf{A}} = \mathbf{A} + \mathbf{I}$ ($\mathbf{I}$ is the identity matrix), and all variables derived from $\tilde{\mathbf{A}}$ are decorated with symbol $\tilde{\ }$, e.g., $\tilde{\mathbf{D}}$ represents the diagonal degree matrix of $\tilde{\mathbf{A}}$. $\mathbf{M}^d$ stands for the $d$-th power of matrix $\mathbf{M}$, while the parameter and node embedding matrices in the $d$-th layer of a GCN are denoted by $\mathbf{W}^{(d)}$ and $\mathbf{H}^{(d)}$.

**Graph convolutional network (GCN) and simplified graph convolutional network (SGC).** The layer-wise message-passing and aggregation of GCN (Kipf & Welling, 2016) is given by

$$\mathbf{H}^{(d+1)} = \sigma(\tilde{\mathbf{D}}^{-\frac{1}{2}} \tilde{\mathbf{A}} \tilde{\mathbf{D}}^{-\frac{1}{2}} \mathbf{H}^{(d)} \mathbf{W}^{(d)}), \tag{1}$$

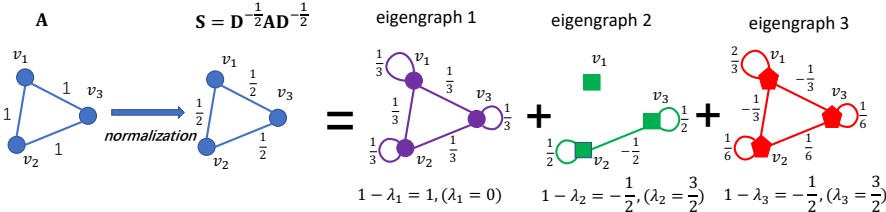

Figure 1: Decompose the symmetrically normalized adjacency matrix into three eigengraphs.

where $\mathbf{H}^{(d)}/\mathbf{H}^{(d+1)}$ stands for the embedding matrix ($\mathbf{H}^{(0)} = \mathbf{X}$) in the $d$-th/$(d+1)$-th layer; $\mathbf{W}^{(d)}$ is the trainable parameter matrix; and $\sigma(\cdot)$ is the non-linear activation function. With $\sigma(\cdot)$ removed in each layer, SGC (Wu et al., 2019) is obtained as below:

$$\mathbf{H}^{(d)} = \tilde{\mathbf{S}}^d \mathbf{X} \mathbf{W}, \qquad (2)$$

where $\tilde{\mathbf{S}} = \tilde{\mathbf{D}}^{-\frac{1}{2}} \tilde{\mathbf{A}} \tilde{\mathbf{D}}^{-\frac{1}{2}}$, and the parameter of each layer $\mathbf{W}^{(i)}$ are compressed into one trainable $\mathbf{W} = \prod_{i=0}^{d-1} \mathbf{W}^{(i)}$.

**Graph Laplacian and spectrum.** In graph theory, graph Laplacian $\mathbf{L} = \mathbf{D} - \mathbf{A}$ and its symmetrically normalized correspondence $\mathbf{L}_{sym} = \mathbf{I} - \mathbf{D}^{-\frac{1}{2}} \mathbf{A} \mathbf{D}^{-\frac{1}{2}}$ possess critical properties of the underlying graph $\mathcal{G}$. $\mathbf{L}_{sym}$ has eigenvalues $[\lambda_1, \lambda_2, \ldots, \lambda_n]$, where $\lambda_i \in [0, 2), \forall i \in \{1, 2, \ldots, n\}$ (Chung & Graham, 1997). [1] Here they are put in ascending order: $0 = \lambda_1 \leq \lambda_2 \leq \cdots \leq \lambda_n < 2$. It can be eigen-decomposed as: $\mathbf{L}_{sym} = \mathbf{U}\mathbf{\Lambda}\mathbf{U}^\top$, where $\mathbf{U} = [\mathbf{u}_1, \mathbf{u}_2, \ldots, \mathbf{u}_n]$ is the eigenvector matrix ($\mathbf{u}_i \perp \mathbf{u}_j, \forall i \neq j$), and $\mathbf{\Lambda}$ is the diagonal eigenvalue matrix:

$$\mathbf{\Lambda} = \begin{bmatrix} \lambda_1 & \cdots & 0 \\ \vdots & \ddots & \vdots \\ 0 & \cdots & \lambda_n \end{bmatrix}.$$

For each eigenvector $\mathbf{u}_i$, we have $\mathbf{u}_i \mathbf{u}_i^\top \in \mathbb{R}^{n \times n}$. As we will show in Section 3, this $n \times n$ matrix can be viewed as a weighted adjacency matrix of a graph with possible negative edges, which we name $\mathbf{u}_i \mathbf{u}_i^\top$ as the $i$-th eigengraph of $\mathcal{G}$. Accordingly, $\mathbf{L}_{sym}$ can be written as the linear combination of all eigengraphs weighted by the corresponding eigenvalues (Chung & Graham, 1997):

$$\mathbf{L}_{sym} = \lambda_1 \mathbf{u}_1 \mathbf{u}_1^\top + \ldots + \lambda_i \mathbf{u}_i \mathbf{u}_i^\top + \ldots + \lambda_n \mathbf{u}_n \mathbf{u}_n^\top, \qquad (3)$$

where the first eigenvalue $\lambda_1 = 0$, and the corresponding eigengraph $\mathbf{u}_1 \mathbf{u}_1^\top$ has an identical value $\frac{1}{n}$ for all entries (Shuman et al., 2013). Thus, for SGC, we have

$$\tilde{\mathbf{S}} = \mathbf{I} - \tilde{\mathbf{L}}_{sym} = \tilde{\mathbf{U}}(\mathbf{I} - \tilde{\mathbf{\Lambda}})\tilde{\mathbf{U}}^\top = \sum_{i=0}^{n} (1 - \tilde{\lambda}_i) \tilde{\mathbf{u}}_i \tilde{\mathbf{u}}_i^\top. \qquad (4)$$

A SGC with $d$ layers requires $d$ consecutive graph convolution operations, which involves the multiplication of $\tilde{\mathbf{S}}$ by $d$ times. Due to the orthogonality of $\tilde{\mathbf{U}}$, namely, $\tilde{\mathbf{U}}^\top \tilde{\mathbf{U}} = \mathbf{I}$, we obtain

$$\tilde{\mathbf{S}}^d = \tilde{\mathbf{U}}(\mathbf{I} - \tilde{\mathbf{\Lambda}})\tilde{\mathbf{U}}^\top \tilde{\mathbf{U}}(\mathbf{I} - \tilde{\mathbf{\Lambda}})\tilde{\mathbf{U}}^\top \ldots \tilde{\mathbf{U}}(\mathbf{I} - \tilde{\mathbf{\Lambda}})\tilde{\mathbf{U}}^\top = \tilde{\mathbf{U}}(\mathbf{I} - \tilde{\mathbf{\Lambda}})^d \tilde{\mathbf{U}}^\top = \sum_{i=1}^{n} (1 - \tilde{\lambda}_i)^d \tilde{\mathbf{u}}_i \tilde{\mathbf{u}}_i^\top, \qquad (5)$$

where $1 - \tilde{\lambda}_i \in (-1, 1]$, and the depth $d$ of SGC serves as the power of $\tilde{\mathbf{S}}$'s eigenvalues. $\tilde{\mathbf{S}}^d$ can be viewed as the sum of eigengraphs $\tilde{\mathbf{u}}_i \tilde{\mathbf{u}}_i^\top$ weighted by coefficients $(1 - \tilde{\lambda}_i)^d$.

**Graph homophily and heterophily.** Graph homophily describes to what extent edges tend to link nodes with the same labels and similar features. In this work, we focus on edge homophily (Zhu et al., 2020): $h(\mathcal{G}) = \frac{\sum_{i,j, \mathbf{A}[i,j]=1} \langle \mathbf{y}[i] = \mathbf{y}[j] \rangle}{\sum_{i,j} \mathbf{A}[i,j]} \in [0, 1]$, where $\langle x \rangle = 1$ if $x$ is true and 0 otherwise. A graph is more homophilic for $h(\mathcal{G})$ closer to 1 or more heterophilic for $h(\mathcal{G})$ closer to 0.

## 3 MODEL

Firstly, we establish the intrinsic connections between eigengraphs with small/large weights, graph signals with high/low frequencies, and graphs with homophilic/heterophilic properties. Secondly, we

---

[1]This work focuses on connected graph without bipartite components (i.e., a connected component which is a bipartite graph).

show how the positive/negative depth $d$ of a GCN affects the eigengraph weights and in turn determines the algorithm's expressive power to process homophilic/heterophilic graph signals. Thirdly, with the help of functional calculus (Shah & Okutmuştur, 2020), we present the theoretical feasibility of extending the domain of $d$ from $\mathbb{N}+$ to $\mathbb{R}$. Finally, by making $d$ a trainable parameter, we present our model RED-GCN and its variants, which are capable of automatically detecting the homophily/heterophily of the input graph and finding the corresponding optimal depth.

The eigenvectors of a graph Laplacian form a complete set of basis vectors in the $n$-dimensional space capable of expressing the original node attribute $\mathbf{X}$ as a linear combination. From the perspective of graph spectrum analysis (Shuman et al., 2013), the frequency of eigenvector $\mathbf{u}_i$ reflects how much the $j$-th entry $\mathbf{u}_i[j]$ deviates from the $k$-th entry $\mathbf{u}_i[k]$ for each connected node pair $v_j$ and $v_k$ in $\mathcal{G}$. This deviation is measured by the set of zero crossings of $\mathbf{u}_i$: $\mathcal{Z}(\mathbf{u}_i) := \{e = (v_j, v_k) \in \mathcal{E} : \mathbf{u}_i[j]\mathbf{u}_i[k] < 0\}$, where $\mathcal{E}$ is the set of edges in graph $\mathcal{G}$. Larger/smaller $|\mathcal{Z}(\mathbf{u}_i)|$ indicates higher/lower eigenvector frequency. A zero-crossing also corresponds a negative weighted edge in an eigengraph. Due to the widely existing positive correlation between $\lambda_i$ and $|\mathcal{Z}(\mathbf{u}_i)|$ (Shuman et al., 2013), large/small eigenvalues mostly correspond to the high/low frequencies of the related eigenvectors. As illustrated by the toy example of $n = 3$ in Figure 1, for $\lambda_1 = 0$, we have $|\mathcal{Z}(\mathbf{u}_1)| = 0$, and eigengraph $\mathbf{u}_1\mathbf{u}_1^\top$ is well-connected with identical edge weight $\frac{1}{n}$; negative edge weights exist in the 2nd and 3rd eigengraphs, indicating more zero crossings ($|\mathcal{Z}(\mathbf{u}_2)| = 1$ and $|\mathcal{Z}(\mathbf{u}_3)| = 2$) and higher eigenvector frequencies.

Since node labels correlate with their attributes (Zheng et al., 2022a), and node attribute similarities indicate the extent of smoothness/homophily (Luan et al., 2020; 2021), plus node attributes can be expressed by eigenvectors, the deviation between eigenvector entry pairs naturally implies the extent of heterophily. Apparently, high frequency eigenvectors and their corresponding eigengraphs have advantage on capturing graph heterophily. High frequency eigengraphs should accordingly take larger weights when modeling heterophilic graphs, while low frequency ones should carry larger weights when dealing with homophilic graphs. In turn, eigengraph weights are controlled by GCN/SGC's depth $d$, e.g., for a SGC of depth $d$, the weight of the $i$-th eigengraph is $(1 - \tilde{\lambda}_i)^d$, and changing the layer $d$ of SGC adjusts the weights of different eigengraphs. Therefore, depth $d$ controls the model's expressive power to effectively filter low/high-frequency signals for graph homophily/heterophily.

A question is naturally raised: instead of manually setting the depth $d$, can $d$ be built into the model as a trainable parameter so that a proper set of the eigengraph weights matching the graph homophily/heterophily can be automatically reached by finding the optimal $d$ in an end-to-end fashion during training? Differentiable variables need continuity, which requires the extension of depth $d$ from the discrete positive integer domain ($\mathbb{N}+$) to the continuous real number domain $\mathbb{R}$. According to functional calculus (Shah & Okutmuştur, 2020), applying an arbitrary function $f$ on a graph Laplacian $\mathbf{L}_{sym}$ is equivalent to applying the same function only on the eigenvalue matrix $\mathbf{\Lambda}$:

$$f(\mathbf{L}_{sym}) = \mathbf{U}f(\mathbf{\Lambda})\mathbf{U}^\top = \mathbf{U} \begin{bmatrix} f(\lambda_1) & \cdots & 0 \\ \vdots & \ddots & \vdots \\ 0 & \cdots & f(\lambda_n) \end{bmatrix} \mathbf{U}^\top, \qquad (6)$$

which also applies to $\tilde{\mathbf{L}}_{sym}$ and $\tilde{\mathbf{S}}$. Armed with this, we seek to realize an *arbitrary* depth SGC via a power function as $f(\tilde{\mathbf{S}}) = \tilde{\mathbf{S}}^d = \tilde{\mathbf{U}}(\mathbf{I} - \tilde{\mathbf{\Lambda}})^d\tilde{\mathbf{U}}^\top = \sum_{i=1}^{n}(1 - \tilde{\lambda}_i)^d\tilde{\mathbf{u}}_i\tilde{\mathbf{u}}_i^\top (d \in \mathbb{R})$.

However, since $\tilde{\lambda}_i \in [0, 2)$, we have $(1 - \tilde{\lambda}_i) \leq 0$ when $1 \leq \tilde{\lambda}_i < 2$, and for $(1 - \tilde{\lambda}_i)$ taking zero or negative values, $(1 - \tilde{\lambda}_i)^d$ is not well-defined or involving complex-number-based calculations for a real-valued $d$ (e.g.,$(-0.5)^{\frac{3}{8}}$) (Shah & Okutmuştur, 2020). Moreover, even for integer-valued $d$s under which $(1 - \tilde{\lambda}_i)^d$ is easy to compute, the behavior of $(1 - \tilde{\lambda}_i)^d$ is complicated versus $\tilde{\lambda}_i$ and diverges when $\tilde{\lambda}_i = 1$ for negative $d$s, as shown in Figure 2a. Thus, the favored weight distribution may be hard to obtain by tuning $d$.

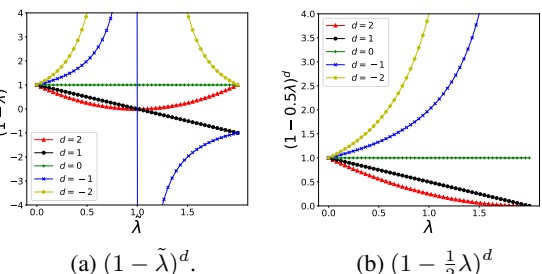

(a) $(1 - \tilde{\lambda})^d$.      (b) $(1 - \frac{1}{2}\lambda)^d$

Figure 2: Eigengraph weight versus eigenvalue for (a) SGC and (b) RED-GCN-S under different depth $d$s.

To avoid such complications and alleviate
the difficulties for manipulating the eigengraph weights, a transformation function $g(\cdot)$ operating on the graph Laplacian $\mathbf{L}_{sym}$ or $\tilde{\mathbf{L}}_{sym}$ is in need to shift $g(\lambda_i)$ or $g(\tilde{\lambda}_i)$ into a proper value range so that its power of a real-valued $d$ is easy to obtain and well-behaved versus $\lambda_i$ or $\tilde{\lambda}_i$. Without the loss of generality, our following analysis focuses on $\mathbf{L}_{sym}$ and $\lambda_i$. There may exist multiple choices for $g(\cdot)$ satisfying the requirements. In this work, we focus on the following form:

$$\hat{\mathbf{S}} = g(\mathbf{L}_{sym}) = \frac{1}{2}(2\mathbf{I} - \mathbf{L}_{sym}). \tag{7}$$

This choice of $g(\cdot)$ holds three properties: (1) *Positive eigenvalues.* Since we have $\mathbf{L}_{sym}$'s $i$-th eigenvalue $\lambda_i \in [0, 2)$, the corresponding eigenvalue of $\hat{\mathbf{S}}$ is $g(\lambda_i) = \frac{1}{2}(2 - \lambda_i) \in (0, 1]$. Thus, the $d$-th power of $g(\lambda_i)$ is computable for any $d \in \mathbb{R}$. (2) *Monotonicity versus eigenvalues $\lambda$.* As shown in Figure 2b, $g(\lambda_i)^d = (1 - \frac{1}{2}\lambda)^d$ is monotonically increasing/decreasing when $\lambda$ varies between 0 and 2 under negative/positive depth. (3) *Geometric interpretability.* Filter $\hat{\mathbf{S}}$ can be expressed as:

$$\hat{\mathbf{S}} = \mathbf{U}\hat{\mathbf{\Lambda}}\mathbf{U}^\top = \mathbf{U}(\mathbf{I} - \frac{1}{2}\mathbf{\Lambda})\mathbf{U}^\top = \frac{1}{2}\mathbf{I} + \frac{1}{2}(\mathbf{I} - \mathbf{L}_{sym}) = \frac{1}{2}(\mathbf{I} + \mathbf{D}^{-\frac{1}{2}}\mathbf{A}\mathbf{D}^{-\frac{1}{2}}). \tag{8}$$

As shown in Figure 3, in spatial domain, $\hat{\mathbf{S}}$ is obtained via 3 operations on adjacency matrix $\mathbf{A}$: normalization, adding self-loops, and scaling all edge weights by $\frac{1}{2}$ (a type of lazy random walk (Luan et al., 2020)), while $\tilde{\mathbf{S}}$ in vanilla GCN/SGC contains 2 operations: adding self-loops and normalization.

With the help of transformation $g$, the depth $d$ is redefined on real number domain, and the message propagation process of depth $d$ can be realized via the following steps: (1) Eigen-decompose $\mathbf{L}_{sym}$; (2) Calculate $\hat{\mathbf{S}}^d$ via weight $g(\lambda_i)^d$ and the weighted sum of all eigengraphs: $\hat{\mathbf{S}}^d = \mathbf{U}\hat{\mathbf{\Lambda}}^d\mathbf{U}^\top = \sum_{i=1}^n g(\lambda_i)^d \mathbf{u}_i \mathbf{u}_i^\top$ (3) Multiply $\hat{\mathbf{S}}^d$ with original node attributes $\mathbf{X}$.

*Negative depth explained.* An intuitive explanation of negative $d$ can be obtained from the perspective of matrix inverse and message diffusion process when $d$ takes integer values. Since $\hat{\mathbf{S}}^{-1}\hat{\mathbf{S}} = \mathbf{U}\hat{\mathbf{\Lambda}}^{-1}\mathbf{U}^\top\mathbf{U}\hat{\mathbf{\Lambda}}^1\mathbf{U}^\top = \mathbf{I}$, $\hat{\mathbf{S}}^{-1}$ is the inverse matrix of $\hat{\mathbf{S}}$. In diffusion dynamics, $\mathbf{X}$ can be viewed as an intermediate state generated in a series of message propagation steps. $\hat{\mathbf{S}}\mathbf{X}$ effectively propagates the message one-step forward, while $\hat{\mathbf{S}}^{-1}$ can cancel the effect of $\hat{\mathbf{S}}$ on $\mathbf{X}$ and recover the original message by moving backward: $\hat{\mathbf{S}}^{-1}\hat{\mathbf{S}}\mathbf{X} = \mathbf{X}$. Accordingly, $\hat{\mathbf{S}}^{-1}\mathbf{X}$

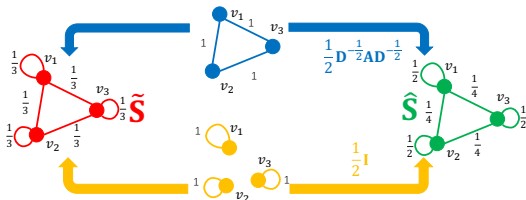

Figure 3: The difference between $\tilde{\mathbf{S}}$ (left) for GCN/SGC and $\hat{\mathbf{S}}$ (right) for RED-GCN.

traces back to the message's previous state in the series. However, neither $\mathbf{A}$ or $\mathbf{L}$ has inverse due to their non-positive eigenvalues. More discussions on the impact of negative depth in spatial domain are presented in Section 4.4. Non-integer $d$ indicates the back- or forward propagation can be a continuous process.

**RED-GCN-S.** By further making $d$ a trainable parameter, we present our model, *Redefined Depth-GCN-Single* (RED-GCN-S), whose final node embedding matrix is given by

$$\mathbf{H} = \sigma(\hat{\mathbf{S}}^d\mathbf{X}\mathbf{W}), \tag{9}$$

where $\sigma(\cdot)$ is the nonlinear activation function; $\mathbf{W}$ is a trainable parameter matrix; and $d$ is the trainable depth parameter. As observed from Figure 2b, weight distribution of different frequencies/eigengraphs is tuned via $d$: (1) for $d = 0$, the weight is uniformed distributed among all frequency components ($g(\lambda_i)^d = 1$), which implies that no particular frequency is preferred by the graph signal; (2) for $d > 0$, weight $g(\lambda_i)^d$ decreases with the corresponding frequency, which indicates the low frequency components are favored so that RED-GCN-S effectively functions as a low-pass filter and therefore captures graph homophily; (3) for $d < 0$, high frequency components gains amplified weights so that RED-GCN-S serves as a high-pass filter capturing graph heterophily. During training, RED-GCN-S tunes its frequency filtering functionality to suit the underlying graph signal by automatically finding the optimal $d$.

**RED-GCN-D.** During optimization, RED-GCN-S embraces a single depth $d$ unified for all eigengraphs and selects its preferences for either homophily or heterophily. However, RED-GCN-S

requires a full eigen-decomposition of $\mathbf{L}_{sym}$, which can be expensive for large graphs. Additionally, the high and low frequency components in a graph signal may not be mutually exclusive, namely, there exists the possibility for a graph to simultaneously possess homophilic and heterophilic counterparts. Therefore, we propose the second variant of RED-GCN: RED-GCN-D (Dual), which introduces two separate trainable depths, $d_h$ and $d_l$, to gain more flexible weighting of the high and low frequency related eigengraphs respectively. Arnoldi method (Lehoucq et al., 1998) is adopted to conduct EVD on $\mathbf{L}_{sym}$ and obtain the top-$K$ largest and smallest eigen-pairs $(\lambda_i, \mathbf{u}_i)$s. By denoting $\mathbf{U}_l = \mathbf{U}[:, 0 : K]$ and $\mathbf{U}_h = \mathbf{U}[:, n - K : n]$ ($\mathbf{U}_l$ and $\mathbf{U}_h \in \mathbb{R}^{n \times K}$), we define a new diffusion matrix $\hat{\mathbf{S}}_{dual}(d_l, d_h, K)$ as

$$\hat{\mathbf{S}}_{dual}(d_l, d_h, K) = \mathbf{U}_l \hat{\mathbf{\Lambda}}_l^{d_l} \mathbf{U}_l^\top + \mathbf{U}_h \hat{\mathbf{\Lambda}}_h^{d_h} \mathbf{U}_h^\top, \tag{10}$$

where $\hat{\mathbf{\Lambda}}_l \in \mathbb{R}^{K \times K}$ and $\hat{\mathbf{\Lambda}}_h \in \mathbb{R}^{K \times K}$ are diagonal matrices of the top-$K$ smallest and largest eigenvalues. [2] The final node embedding of RED-GCN-D is presented as

$$\mathbf{H} = \sigma(\hat{\mathbf{S}}_{dual}(d_l, d_h, K)\mathbf{X}\mathbf{W}), \tag{11}$$

where depths $d_l$ and $d_h$ are trainable; and $\mathbf{W}$ is a trainable parameter matrix. We make RED-GCN-D scalable on large graphs by choosing $K \ll n$, so that $\hat{\mathbf{S}}_{dual}(d_l, d_h, K)$ approximates the full diffusion matrix by covering only a small subset of all eigengraphs. For small graphs, we use $K = \lfloor \frac{n}{2} \rfloor$ to include all eigengraphs, and $\hat{\mathbf{S}}_{dual}(d_l, d_h, K)$ thus gains higher flexibility than $\hat{\mathbf{S}}$ with the help of the two separate depth parameters instead of a unified one.

*Differences with ODE-based GNNs.* Previous attempts on GNNs with continuous diffusion are mostly inspired by graph diffusion equation, an Ordinary Difference Equation (ODE) characterizing the dynamical message propagation process versus time. In contrast, our framework starts from discrete graph convolution operations without explicitly involving ODE. CGNN (Xhonneux et al., 2020) aims to build a deep GNN immune to over-smoothing by adopting the neural ODE framework (Chen et al., 2018). But its time parameter $t$ is a non-trainable hyper-parameter predefined within the positive domain, which is the key difference with RED-GCN. A critical CGNN component for preventing over-smoothing, restart distribution (the skip connection from the first layer), is not needed in our framework. Moreover, CGNN applies the same depth to all frequency components, while RED-GCN-D has the flexibility to adopt two different depths respectively to be adaptive to high and low frequency components. GRAND (Chamberlain et al., 2021) introduces non-Eular multi-step schemes with adaptive step size to obtain more precise solutions of the diffusion equation. Its depth (total integration time) is continuous but still predefined/non-trainable and takes only positive values. DGC (Wang et al., 2021) decouples the SGC depth into two prefinded non-trainable hyper-parameters: a positive real-valued $T$ controlling the total time and a positive integer-valued $K_{dgc}$ corresponding to the number of diffusion steps. However, realizing negative depth in DGC is non-applicable since the implementation is through propagation by $K_{dgc}$ times, rather than through an arbitrary real-valued exponent $d$ on eigengraph weights in RED-GCN.

## 4 EXPERIMENT

In this section, we evaluate the proposed RED-GCN on the semi-supervised node classification task on both homophilic graphs and heterophilic graphs.

### 4.1 EXPERIMENT SETUP

**Datasets.** We use 11 datasets for evaluation, including 4 homophilic graphs: Cora (Kipf & Welling, 2016), Citeseer (Kipf & Welling, 2016), Pubmed (Kipf & Welling, 2016) and DBLP (Bojchevski & Günnemann, 2017), and 7 heterophilic graphs: Cornell (Pei et al., 2020), Texas (Pei et al., 2020), Wisconsin (Pei et al., 2020), Actor (Pei et al., 2020), Chameleon (Rozemberczki et al., 2021), Squirrel (Rozemberczki et al., 2021), and cornell5 (Fey & Lenssen, 2019). We collect all datasets from the public GCN platform Pytorch-Geometric (Fey & Lenssen, 2019). For Cora, Citeseer, Pubmed with data splits in Pytorch-Geometric, we keep the training/validation/testing set split as in GCN (Kipf & Welling, 2016). For the remaining 8 datasets, we randomly split every dataset into 20/20/60% for training, validation, and testing. The statistics of all datasets are presented in Appendix.

---

[2]Case $\lambda_i = 2$, namely $g(\lambda_i) = 0$, is excluded since it corresponds to the existence of bipartite components.

**Baselines and Metrics.** We compare our model with 7 baseline methods,including 4 classic GNNs: GCN (Kipf & Welling, 2016), SGC (Wu et al., 2019), APPNP (Klicpera et al., 2018) and ChebNet (Defferrard et al., 2016), and 3 GNNs tailored for heterophilic graphs: FAGCN (Bo et al., 2021), GPRGNN (Chien et al., 2020) and H2GCN (Zhu et al., 2020). Accuracy (ACC) is used as the evaluation metric. We report the average ACCs with the standard deviation (std) for all methods, each obtained by 5 runs with different initializations.

**Implementation Details.** See Appendix due to the page limit.

## 4.2 Node Classification

The semi-supervised node classification performances on homophilic graphs and heterophilic graphs are shown in Table 1 and Table 2 respectively.

*Homophilic graphs*. From Table 1, it is observed that different methods have similar performance on homophilic graphs. RED-GCN-S achieves the best accuracies on two datasets: Cora and DBLP. On the remaining two datasets, RED-GCN-S is only $1.1\%$ and $0.4\%$ below the best baselines (APPNP on Citeseer and SGC on Pubmed). For RED-GCN-D, it obtains similar performance as the other methods, even though it only uses the top-$K$ largest/smallest eigen-pairs.

Table 1: Performance comparison (mean±std accuracy) on homophilic graphs.

| Datasets | Cora | Citeseer | Pubmed | DBLP |
|---|---|---|---|---|
| GCN | 80.8±0.8 | 70.5±0.6 | 78.8±0.6 | 84.1±0.2 |
| SGC | 80.9±0.4 | 70.8±0.8 | **79.6±0.4** | 84.1±0.2 |
| APPNP | 81.0±1.0 | **71.9±0.4** | 79.3±0.2 | 83.0±0.5 |
| GPRGNN | 82.0±0.7 | 69.3±0.9 | 78.6±0.7 | 84.5±0.3 |
| FAGCN | 80.3±0.4 | 71.7±0.8 | 78.5±0.9 | 82.4±0.7 |
| H2GCN | 78.8±1.0 | 70.5±1.0 | 77.9±0.3 | 82.4±0.3 |
| ChebNet | 78.8±0.5 | 71.1±0.4 | 78.1±0.8 | 83.1±0.1 |
| RED-GCN-S | **82.5±1.1** | 70.8±0.7 | 79.2±0.2 | **84.7±0.3** |
| RED-GCN-D | 82.4 ±0.7 | 70.6 ±0.6 | 77.9 ±0.3 | 84.2±0.2 |

Table 2: Performance comparison (mean±std accuracy) on heterphilic graphs.

| Datasets | Texas | Cornell | Wisconsin | Actor | Squirrel | Chameleon | cornell5 |
|---|---|---|---|---|---|---|---|
| GCN | 55.9±3.4 | 44.3±4.4 | 51.4±2.2 | 27.5±0.5 | 35.8±1.3 | 55.2±1.8 | 67.9±0.2 |
| SGC | 58.7±3.1 | 43.8±4.4 | 47.3±2.1 | 28.0±0.8 | 37.2±1.8 | 55.3±1.0 | 67.4±0.5 |
| APPNP | 55.1±3.7 | 51.5±2.4 | 58.0±3.1 | 32.8±0.8 | 29.5±0.9 | 46.7±0.8 | 68.3±0.5 |
| GPRGNN | 61.3 ±5.8 | 53.3±4.6 | 71.0±4.8 | 33.6±0.4 | 34.1±1.0 | 55.0±3.9 | 67.3±0.3 |
| FAGCN | 60.2±7.8 | 54.8±7.4 | 60.1±5.2 | 32.3±0.5 | 31.2±1.6 | 50.4±1.9 | 68.3±0.7 |
| H2GCN | 68.8±6.5 | 61.4±4.4 | 69.9±5.3 | 33.9±0.3 | 30.4±0.9 | 48.8±1.9 | 68.4±0.2 |
| ChebNet | 76.2±2.9 | 66.7±3.9 | 75.4±3.5 | 34.3±0.5 | 31.8±0.5 | 49.6±1.8 | OOM |
| RED-GCN-S | **77.6±5.9** | 72.0±5.8 | **82.0±2.6** | **35.3±0.7** | 38.2±1.2 | 55.7±1.3 | 68.5±0.4 |
| RED-GCN-D | 77.1 ±2.5 | **72.0 ±2.8** | 81.5 ±2.4 | 27.6 ±0.8 | **44.2±0.9** | **56.9±0.9** | **70.0±0.2** |

*Heterophilic graphs*. RED-GCN-S/RED-GCN-D outperforms every baseline on all heterophilic graphs, as shown in Table 2. These results demonstrate that without manually setting the model depth and without the prior knowledge of the input graph, RED-GCN has the capability of automatically detecting the underlying graph heterophily. We have an interesting observation: on 3 large datasets, Squirrel, Chameleon, and cornell5, even with only a small portion of the eigengraphs, RED-GCN-D is able to achieve better performance than RED-GCN-S with the complete set of eigengraphs. This suggests that the graph signal in some real-world graphs might be dominated by a few low and high frequency components, and allowing two independent depth parameters in RED-GCN-D brings the flexibility to capture the low and high frequencies at the same time.

## 4.3 trainable depth

A systematic study is conducted on the node classification performance w.r.t the trainable depth $d$.

*Optimal depth*. In Figure 4, the optimal depths and their corresponding classification accuracies are annotated. For two homophilic graphs, Cora and Citeseer, the optimal depths are positive (5.029 and 3.735) in terms of the best ACCs, while for two heterophilic graphs, Actor and Squirrel, the optimal depths are negative ($-0.027$ and $-3.751$). These results demonstrate our model indeed automatically capture graph heterophily/homophily by finding the suitable depth to suppress or amplify the relative weights of the corresponding frequency components. Namely, high/low frequency components are suppressed for homophilic/heterophilic graphs respectively.

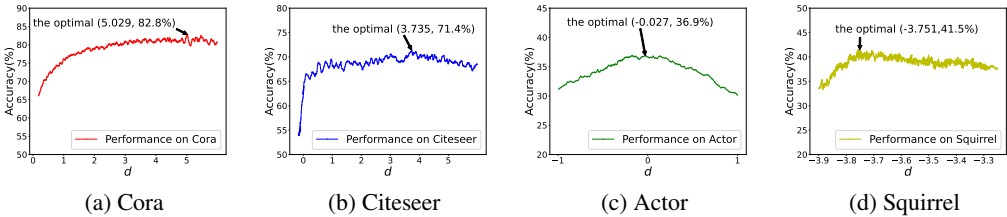

| (a) Cora | (b) Citeseer | (c) Actor | (d) Squirrel |

Figure 4: Node classification accuracy w.r.t. the trainable depth $d$ on four datasets: Cora, Citeseer, Actor and Squirrel. (the optimal $d$, accuracy) is annotated (e.g., (-0.027, 36.9%) for Actor).

*Close to zero depth.* For the two homophilic graphs in Figures 4a and 4b, sharp performance drop is observed when depth $d$ approaches 0, since the eigengraphs gain close-to-uniform weights. For the heterophilic Actor dataset, its optimal depth $-0.027$ is close to 0, as shown in Figure 4c. In addition, the performance of RED-GCN-D (27.6%) is similar to that of GCN (27.5%), both of which are much worse than RED-GCN-S (35.3%). This result indicates that Actor is a special graph where all frequency components have similar importance. Due to the absence of the intermediate frequency components between the high- and low-end ones, the performance of RED-GCN-D is severely impacted. For vanilla GCN, the suppressed weights of the high frequency components deviate from the near-uniform spectrum and thus lead to low ACC on this dataset.

### 4.4 GRAPH AUGMENTATION AND GEOMETRIC INSIGHTS

It is especially interesting to analyze what change a negative depth brings to the spatial domain and how such change impacts the subsequent model performance. *Graph augmentation.* By picking the optimal depth $d$ according to the best performance on the validation set, a new diffusion matrix $\hat{\mathbf{S}}^d$ is obtained. With the optimal $d$ fixed, substituting the normalized adjacency matrix $\tilde{\mathbf{D}}^{-\frac{1}{2}} \tilde{\mathbf{A}} \tilde{\mathbf{D}}^{-\frac{1}{2}}$ in Eq. 1 by $\hat{\mathbf{S}}^d$ is equivalent to applying the vanilla GCN to a new topology. This topology effectively plays the role

Table 3: The performance of one-layer vanilla GCN over the augmented $\hat{\mathbf{S}}^d$.

| Datasets | Texas | Cornell | Wisconsin |
|---|---|---|---|
| GCN | 55.9 | 44.3 | 51.4 |
| RED-GCN-S | **77.6** | 72.0 | 82.0 |
| RED-GCN-D | 77.1 | 72.0 | 81.5 |
| GCN ($\hat{\mathbf{S}}^d$) | 75.9 | **72.7** | **83.4** |

of a structural augmentation for the original graph. The impact of such augmentation on performance is tested on 3 heterophilic graphs: Texas, Cornell and Wisconsin, as shown in Table 3. Apparently, for the vanilla GCN, the performance obtained with this new topology is superior over that with the raw input graph: it dramatically brings 20%-30% lifts in ACC. Moreover, the augmented topologies also make vanilla GCN outperform RED-GCN-S and RED-GCN-D on 2 out of the 3 datasets. By nature, the augmented graph is a re-weighted linear combination of the eigengraphs, and its topological structure intrinsically assigns higher weights to eigengraphs corresponding to higher frequencies, as shown in Figures 5.

*Geometric properties.* To further understand how the topology of $\hat{\mathbf{S}}^d$ with a negative optimal $d$ differs from that of $\hat{\mathbf{S}}$ and why the performance is significantly boosted, a heat map of $(\hat{\mathbf{S}}^d - \hat{\mathbf{S}})$ is presented in Figure 6 for Cornell. [3] First, the dark red diagonal line in the heat map indicates the weights of self-loops are significantly strengthened in the augmented graph, and as a result, in consistency with the previous findings (Zheng et al., 2022a), the raw node attributes make more contributions in determining their labels. These strengthened self-weights also play the similar role as restart distribution or skip connections (Xhonneux et al., 2020) preventing the node embeddings becoming over-smoothed. In addition, there is a horizontal line and a vertical line (light yellow line marked by dashed ovals) in the heat map in Figure 6, correspond to the hub node in the graph, namely the node with the largest degree. Interestingly, the connections between this node and most other nodes in the graph experience a negative weight change. Therefore, the influence of the

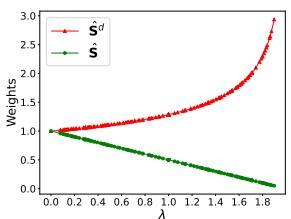

Figure 5: The weights of eigengraphs w.r.t. eigenvalues on the augmented diffusion matrix $\hat{\mathbf{S}}^d$ and original $\hat{\mathbf{S}}$ for Cornell ($d = -0.362$).

---

[3]Heat maps for Texas and Wisconsin are in Appendix with similar observations.

hub node on most other nodes are systematically reduced. Consequently, the augmentation amplifies the deviations between node embeddings and facilitates the characterization of graph heterophily.

## 5 RELATED WORKS

**Graph Convolutional Network (GCN).** GCN models can be mainly divided into two categories: (1) spectral graph convolutional networks and (2) spatial convolutional networks. In (1), Spectral CNN (Bruna et al., 2013) borrows the idea from convolutional neural network (Goodfellow et al., 2016) to construct a diagonal matrix as the convolution kernel. ChebNet (Defferrard et al., 2016) adopts a polynomial approximation of the convolution kernel. GCN (Kipf & Welling, 2016) further simplifies the ChebNet via the first order approximation. Recently, (He et al., 2021; Bianchi et al., 2021; Wang & Zhang, 2022) propose more advanced filters as the convolution kernel. Most works in (2) follow the *message-passing* mechanism. GraphSAGE (Hamilton et al., 2017) iteratively aggregates features from local neighborhood. GAT (Veličković et al., 2017) applies self-attention to the neighbors. APPNP (Klicpera et al., 2018) deploys personalized pagerank (Tong et al., 2006) to sample nodes for aggregation. MoNet (Monti et al., 2017) unifies GCNs in the spatial domain.

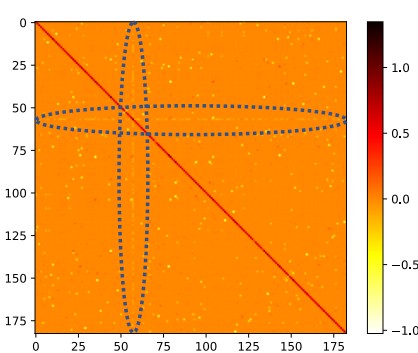

Figure 6: The difference between the augmented diffusion matrix and the original one $\hat{\mathbf{S}}^d - \hat{\mathbf{S}}$ for Cornell in heat map. Best viewed in color.

**The Depth of GCN and Over-smoothing.** A large amount of works focus on the over-smoothing issue. Its intrinsic cause is demystified: a linear GCN layer is a Laplacian smoothing operator (Li et al., 2018; Wu et al., 2019). PairNorm (Zhao & Akoglu, 2019) forces distant nodes to be distinctive by adding an intermediate normalization layer. Dropedge (Rong et al., 2019), DeepGCN (Li et al., 2019), AS-GCN (Huang et al., 2018), and JK-net (Xu et al., 2018) borrow the idea of ResNet (He et al., 2016) to dis-intensify smoothing. DeeperGXX (Zheng et al., 2021) adopts a topology-guided graph contrastive loss for connected node pairs to obtain discriminative representations. Most works aim to build deep GCNs (i.e., $d$ is a large positive integer) by reducing over-smoothing, while RED-GCN extends the depth from $\mathbb{N}+$ to $\mathbb{R}$ and explores the negative depth.

**Node Classification on Homophilic and Heterophilic Graphs.** GCN/GNN models mostly follow the homophily assumption that connected nodes tend to share similar labels (Kipf & Welling, 2016; Veličković et al., 2017; Hamilton et al., 2017). Recently, heterophilic graphs, in which neighbors often have disparate labels, attract lots of attention. Geom-GCN (Pei et al., 2020) and H2GCN (Zhu et al., 2020) extend the neighborhood for aggregation. FAGCN (Bo et al., 2021) and GPRGNN (Chien et al., 2020) adaptively integrate the high/low frequency signals with trainable parameters. Alternative message-passing mechanisms have been proposed in HOG-GCN (Wang & Zhang, 2022) and CPGNN (Zhu et al., 2021). The latest related works include ACM-GCN (Luan et al., 2021; 2022b), LINKX (Lim et al., 2021), BernNet (He et al., 2021), GloGNN (Li et al., 2022) and GBKGNN (Du et al., 2022). Other works can be found in a recent survey (Zheng et al., 2022b).

## 6 CONCLUSION AND FUTURE WORK

To our best knowledge, this work presents the first effort to make GCN's depth trainable by redefining it on the real number domain. We unveil the interdependence between negative GCN depth and graph heterophily. A novel problem of automatic GCN depth tuning for graph homophily/heterophily detection is formulated, and we propose a simple and powerful solution named RED-GCN with two variants (RED-GCN-S and RED-GCN-D). An effective graph augmentation method is also discussed via the new understanding on the message propagation mechanism generated by the negative depth. Superior performance of our method is demonstrated via extensive experiments with semi-supervised node classification on 11 graph datasets. The new insights on GCN's depth obtained by our work may open a new direction for future research on spectral and spatial GNNs. Since RED-GCN requires to conduct eigen-decomposition of the graph Laplacian, it is not directly applicable to inductive and dynamic graph learning problems, which we leave for future exploration.

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
