# OpenReview forum: "ReD-GCN: Revisit the Depth of Graph Convolutional Network"
_ICLR.cc/2023/Conference — Submitted to ICLR 2023_

### Official Review · Reviewer_Qq6e · 2022-10-24

**Confidence:** 4
**Correctness:** 3
**Technical Novelty And Significance:** 3
**Empirical Novelty And Significance:** 3
**Recommendation:** 6

**Clarity, Quality, Novelty And Reproducibility:**

Clarity: overall the paper is well-organized but some parts are hard to follow, e.g., how to learn the depth d is not introduced very clearly.

Quality: the work is of average quality.

Novelty: The proposed method to extend the depth of GNNs from the positive integers to real values is novel.

Reproducibility: it should be possible to implement the method and reproduce the results from the description given in the paper.

**Strength And Weaknesses:**

Strength:
- Both problems of GNN depth and adaptive GNN for modeling homophily and heterophily are interesting and important.
- The proposed method to extend positive integer depth to real value is novel.
- The experimental results of the method demonstrate the effectiveness of the proposed method compared to previous baselines.

Weakness:
- The claims are based on empirical results and theoretical analysis is not provided.
- Experimental results demonstrate the effectiveness of the proposed method, it is also important to show the efficiency of the method, e..g, presenting the complexity or comparing running time.
- More insights can be provided to better show the correlations between homophily/heterophily and positive/negative depth, for example, analyzing the learn representations rather than just reporting the overall classification accuracy in the downstream task.
- The size of the benchmark dataset is relatively small to show the ability of generalization. I suggest conducting experiments on larger datasets such as OGB.
- In the experimental studies, some representative methods can be compared including (1) GAT, as an attention method, and (2) some recent work on adaptive GNN for modeling homophily and heterophily, for instance, GBK-GNN [1].


[1] GBK-GNN: Gated Bi-Kernel Graph Neural Networks for Modeling Both Homophily and Heterophily

**Summary Of The Paper:**

This paper studies the problem of investigating the depth of GNN. The authors propose a novel and interesting method to extend the depth of GNNs from a positive integer to a real value. It is claimed that negative depth enables high-pass frequency filtering functionality for graph heterophily while positive value enables low-pass filter and is to model homophily. Experimental studies have been conducted on both homophilous and heterophilous graphs and the results show that the proposed method can achieve comparable or better performance in both types of graphs compared to baselines.

**Summary Of The Review:**

This paper studies an interesting and important problem. Overall this paper is well-organized. Experiments from different aspects have been conducted. Although experimental studies can be further improved by providing more insight into investigations and comparing the proposed method to more representative methods, the method is novel from the methodological perspective and effective from the experimental perspective. Thus, I will recommend a weak acceptance of the paper.

---

> ### Author Response · Authors · 2022-11-15
> **Response to Reviewer Qq6e**
>
> **Weakness 1**: The claims are based on empirical results and theoretical analysis is not provided.
>
> **Answer**: Our work starts from rigorous theory of graph spectrum [1] [2]. With this solid theoretical foundation on graph eigen-decomposition, the connection between graph signal frequency, zero crossing of eigenvectors, and graph homophily/heterophily is naturally built. Our contribution mainly lies in utilizing these theoretical findings in a novel way to redefine GCN's depth and adjust its frequency profile [2]. In summary, the methodology is initiated by solid theory, so further theoretical analysis on empirical results is omitted.
>
> [1] David I Shuman, Sunil K Narang, Pascal Frossard, Antonio Ortega, and Pierre Vandergheynst. The emerging field of signal processing on graphs: Extending high-dimensional data analysis to networks and other irregular domains. IEEE signal processing magazine, 30(3):83–98, 2013.
>
> [2] Muhammet Balcilar, Guillaume Renton, Pierre He ́roux, Benoit Gau ̈ze`re, Se ́bastien Adam, Paul Honein.Analyzing the Expressive Power of Graph Neural Networks in a Spectral Perspective. ICLR 2021.
>
> **Weakness 2**: Experimental results demonstrate the effectiveness of the proposed method, it is also important to show the efficiency of the method, e..g, presenting the complexity or comparing running time.
>
> **Answer**: We provide a brief complexity analysis of ReDGCN here. ReDGCN has two major steps: (1) conduct eigen-decomposition of $\hat{\mathbf{A}}$; (2) train a one-layer GCN on the augmented graph $\hat{\mathbf{S}}^{d}$. For Step (1), naive eigen-decompostion has a time complexity of O($n^3$). However, by adopting the Lanczos method [1] and only keeping the top-$K$ largest/smallest eigenvalues in ReDGCN-D, the time complexity can be reduced to $O(nK^2 + mK)$ [2], where $m$ is the number of edges and $K<<n$. For Step (2), via dense matrix multiplications, computing the approximated $\hat{\mathbf{S}}^{d}$ takes O($nK^2 + n^{2}K$), and conducting message propagation takes O($n^2q + nqc$), where $q$ and $c$ denote the dimension of node feature vectors and the number of node classes. In our vanilla implementation on the PubMed dataset (19717 nodes and 44338 edges), Step (1) takes 187 seconds, while each epoch in Step (2) takes 0.28 seconds.
>
> Potential improvements of efficiency: since the diffusion matrix of the augmented graph is sparse for known real-world graphs, the complexity of Step (2) can also be further simplified: the O($n^2K$) term can be reduced to O($|E_{aug}|$ K), where $E_{aug}$ is the edge set of the augmented graph.
>
> The key contribution of our paper is the discovery of the correlation between GCN's depth and graph heterophily/homophily. We will optimize/accelerate our vanilla implementation by utilizing sparse matrix computation and other relevant techniques in future works. Thanks again for the constructive suggestion.
>
> [1] Richard B Lehoucq, Danny C Sorensen, and Chao Yang. 1998. ARPACK users’ guide: solution of large-scale eigenvalue problems with implicitly restarted Arnoldi methods. SIAM.
> [2] Zexi Huang, Arlei Silva, and Ambug Singh. A Broader Picture of Random-walk Based Graph Embedding. KDD'2021.

---

> > ### Author Response · Authors · 2022-11-15
> > **Response to Reviewer Qq6e (Continue)**
> >
> > **Weakness 3**: More insights can be provided to better show the correlations between homophily/heterophily and positive/negative depth, for example, analyzing the learn representations rather than just reporting the overall classification accuracy in the downstream task.
> >
> > **Answer**: Thanks for the suggestion. To better show the correlations between homophily/heterophily and positive/negative depth, we provide the insights from two aspects:
> >
> > (1) Following the reviewer's suggestions, we include a visualization of the learned node embeddings under different depths on Cora (homophilic) and Texas (heterophilic) in the appendix of the revised version. **Please download the revised version of the supplementary materials and check the Appendix.** As shown in Figure 3,  for Cora, when $d=5$, the nodes in different classes form clearly distinguishable clusters (ACC=82.4\%). When the depth decreases from a positive value $d=5$ (ACC=82.4\%) to a negative value $d=-2$ (ACC=29.5\%), the nodes belonged to different classes/colors mix with each other, and the clearly clustered structure does not exist anymore. However, for Texas shown in Figure 4, when decreasing the depth from $d=4$ (ACC=50\%) to $d=-0.35$ (ACC=80\%), the clusters of different classes/colors become well-separated. Especially, the purple nodes spread randomly all over the embedding space when $d=4$ but concentrate in one compact cluster when $d=-0.35$. The spatial distribution of the node embeddings reveals that ReDGCN with positive and negative depths respectively models graph homophily and heterophily in a proper way.
> >
> > (2) The augmented graph topology, as explained in *Geometric Properties* in Section 4.4. ``To further understand how the topology of $\hat{\mathbf{S}}^d$ with a negative optimal $d$ differs from that of $\hat{\mathbf{S}}$ and why the performance is significantly boosted, a heat map of $(\hat{\mathbf{S}}^d-\hat{\mathbf{S}})$ is presented in Figure 6 for Cornell. First, the dark red diagonal line in the heat map indicates the weights of self-loops are significantly strengthened in the augmented graph, which prevents the node embeddings becoming over-smoothed. In addition, there is a horizontal line and a vertical line (light yellow line marked by dashed ovals) in the heat map in Figure 6, correspond to the hub node in the graph, namely the node with the largest degree. Interestingly, the connections between this node and most other nodes in the graph experience a negative weight change. Therefore, the influence of the hub node on most other nodes are systematically reduced.''

---

> > > ### Author Response · Authors · 2022-11-15
> > > **Response to Reviewer Qq6e (Continue)**
> > >
> > > **Weakness 4**: The size of the benchmark dataset is relatively small to show the ability of generalization. I suggest conducting experiments on larger datasets such as OGB.
> > >
> > > **Answer**: Thanks for the suggestion. Following reviewer's suggestion, we have conducted additional experiments on the OGB-arxiv  dataset (169,343 nodes and 1,166,243 edges). The performances of different methods on this large dataset under a 20\%/20\%/60\% training/validation/testing split are presented in Table 2. It is observed that ReDGCN still outperforms all the remaining methods on the OGB-arxiv dataset. This further verifies the generalization ability of our proposed model.
> > >
> > > Performance comparison (mean $\pm$ std accuracy) on OGB-arxiv.
> > > |Methods | GCN | SGC | APPNP | GAT | FAGCN |GPRGNN| H2GCN | ChebGCN | ReDGCN|
> > > | ---- | ---- | ---- | ---- | ---- | ---- | ----|---- | ---- | ---- |
> > > |Accuracy | $51.9\pm 0.7$ | $ 57.9 \pm 0.1$ | $55.9\pm 1.0$ | $57.8\pm 0.1$ | $57.5\pm 0.5$| $47.3\pm 0.9$ |  $51.7\pm 2.4$ | $54.2 \pm 0.1$|$\mathbf{63.1} \pm\mathbf{0.7}$|

---

> > > > ### Author Response · Authors · 2022-11-15
> > > > **Response to Reviewer Qq6e (Continue)**
> > > >
> > > > **Weakness 5**: In the experimental studies, some representative methods can be compared including (1) GAT, as an attention method, and (2) some recent work on adaptive GNN for modeling homophily and heterophily, for instance, GBK-GNN [1].
> > > >  [1] GBK-GNN: Gated Bi-Kernel Graph Neural Networks for Modeling Both Homophily and Heterophily.
> > > >
> > > > **Answer**: The authors of GBK-GNN report the performances of GAT and GBK-GNN on four datasets: Cora, Citeseer, Cornell and Texas under a 20\%/20\%/60\% training/validation/testing split, same with our paper. So we compare their reported performances of GAT and GBK-GNN with ReDGCN for these four datasets, as shown in Table 3. For homophilic graph like Cora and Citeseer, ReDGCN has close performance with GBK-GNN. However, for the two heterophic graphs (Cornell and Texas), ReDGCN performs much better than GAT and GBK-GNN, which validates the effectiveness of the proposed ReDGCN.
> > > >
> > > > We add the GAT and GBK-GNN literature into the references of the revised version of our paper.
> > > > Performance comparison on Cora, Citeseer Cornell and Texas.
> > > > |Datasets | Cora | Citeseer | Cornell | Texas|
> > > > | ---- | ---- | ---- | ---- | ---- |
> > > > GAT | $79.91$ | $68.89$ | $65.79$ | $59.21$|
> > > > |GBKGNN | $\underline{84.68}$ | $\underline{72.87}$ | $67.90$ | $70.49$|
> > > > |ReDGCN-S | $\mathbf{84.9}$ | $72.39$ | $\underline{72.0}$ | $\textbf{77.6}$|
> > > > ReDGCN-D | $83.6$ | $\mathbf{74.2}$ | $\mathbf{72.7}$|$\underline{77.1}$|

---

> > > > > ### Comment · Reviewer_Qq6e · 2022-11-21
> > > > > **Response to Authors**
> > > > >
> > > > > Thanks for your response with discussions and more experimental results. I will keep my score but I definitely appreciate your effort, which has positively impacted my impression of the paper.

---

> > > > > > ### Author Response · Authors · 2022-11-22
> > > > > > **Thanks to Reviewer Qq6e**
> > > > > >
> > > > > > Thanks for your reply and your appreciation of our paper !!!

---

### Official Review · Reviewer_NDJu · 2022-10-25

**Confidence:** 4
**Correctness:** 3
**Technical Novelty And Significance:** 3
**Empirical Novelty And Significance:** 3
**Recommendation:** 6

**Clarity, Quality, Novelty And Reproducibility:**

quality: medium
clarity: mediem
originality of the work: medium
Reproducibility: NA

**Strength And Weaknesses:**

## Strength
The idea of setting the depth as trainable continuous value and the concept of eigengraph are interesting.

## Weaknesses
1. The writing needs to be improved.
2. Some experiments and comparisons are missing.


## Questions and Comments

1. “Apparently, high frequency eigenvectors and their corresponding eigengraphs have advantage on capturing graph heterophily” This is not apparent, need more explanation.

2. “High frequency eigengraphs should accordingly take larger.  weights when modeling heterophilic graphs, while low frequency ones should carry larger weights when dealing with homophilic graphs.” This is similar as [5].

3. Equation (7) is actually lazy random walk affinity matrix and see [5] for the generalized lazy random walk matrix.

4. Red-GCN is essentially a linear model without any non-linearity?

5. The reported results of FAGCN, GPRGNN and H2GCN in table 2 are not the same as the results in the original papers and not consistent with my personal experience.

6. Some missing comparisons, e.g. ACM-GCN [1], LINKX [2], BernNet [3] and GloGNN [4].

[1] Luan S, Hua C, Lu Q, et al. Is Heterophily A Real Nightmare For Graph Neural Networks To Do Node Classification?[J]. arXiv preprint arXiv:2109.05641, 2021.

[2] Lim D, Hohne F, Li X, et al. Large scale learning on non-homophilous graphs: New benchmarks and strong simple methods[J]. Advances in Neural Information Processing Systems, 2021, 34: 20887-20902

[3]  He M, Wei Z, Xu H. Bernnet: Learning arbitrary graph spectral filters via bernstein approximation[J]. Advances in Neural Information Processing Systems, 2021, 34: 14239-14251.

[4] Li X, Zhu R, Cheng Y, et al. Finding Global Homophily in Graph Neural Networks When Meeting Heterophily[J]. arXiv preprint arXiv:2205.07308, 2022.

[5] Luan S, Zhao M, Hua C, et al. Complete the missing half: Augmenting aggregation filtering with diversification for graph convolutional networks[J]. arXiv preprint arXiv:2008.08844, 2020.


**Summary Of The Paper:**

The authors define GCN’s depth as a trainable continuous parameter within (−∞,+∞) and propose RED-GCN which can automatically search for the optimal depth without the prior knowledge regarding whether the input graph is homophilic or heterophilic.

**Summary Of The Review:**

This is a borderline paper and if the authors can address my concerns satisfactorily, I will consider raise my score.

---

> ### Author Response · Authors · 2022-11-14
> **Response to Reviewer NDJu**
>
> **Question 1**: “Apparently, high frequency eigenvectors and their corresponding eigengraphs have advantage on capturing graph heterophily” This is not apparent, need more explanation.
>
> **Answer**:  Thanks for the comment. The entries of the eigenvector can be treated as the attributes of the corresponding eigengraph nodes (as distinguished from graph node attributes). Then the edge weight $\mathbf{u}_i(j) \mathbf{u}_i(k)$ of eigengraph $\mathbf{u}_i\mathbf{u}_i^{\top}$ can measure similarities between the eigengraph nodes $j$ and $k$: positive/negative weight represents that $j$ and $k$ are similar/dissimilar. This is also consistent with the definition of zero crossing [1]: number of negative edge weights determines the frequency of the eigengraph/eigenvector. Since node labels correlate with their attributes [2], node attribute similarities naturally imply the extent of homophily or smoothness [3] [4]. Thus, the number of negative eigengraph edge weights reveals the connection between high frequency, node dissimilarity, and graph heterophily. As a result, high frequency eigenvectors and their corresponding eigengraphs have advantage on capturing graph heterophily.
> We have added more explanation into the revised version.
>
> [1] David I Shuman, Sunil K Narang, Pascal Frossard, Antonio Ortega, and Pierre Vandergheynst. The emerging field of signal processing on graphs: Extending high-dimensional data analysis to networks and other irregular domains. IEEE signal processing magazine, 30(3):83–98, 2013.
>
> [2] Wenqing Zheng, W Edward Huang, Nikhil Rao, Sumeet Katariya, Zhangyang Wang, and Karthik Subbian. Cold brew: Distilling graph node representations with incomplete or missing neighborhoods. In International Conference on Learning Representations, 2022.
>
> [3] Luan S, Zhao M, Hua C, et al. Complete the missing half: Augmenting aggregation filtering with diversification for graph convolutional networks[J]. arXiv preprint arXiv:2008.08844, 2020.
>
> [4] Luan S, Hua C, Lu Q, et al. Is Heterophily A Real Nightmare For Graph Neural Networks To Do Node Classification?[J]. arXiv preprint arXiv:2109.05641, 2021.

---

> > ### Author Response · Authors · 2022-11-15
> > **Response to Reviewer NDJu (Continue)**
> >
> > **Question 2**: “High frequency eigengraphs should accordingly take larger weights when modeling heterophilic graphs, while low frequency ones should carry larger weights when dealing with homophilic graphs.” This is similar as [5].
> >
> > **Answer**: The reviewer is right that our argument is consistent with [5]. Reference [5] has been added into the previous sentence in the revised paper as a citation to further support the argument. Differing from [5], our work further reveals the correlation between GCN's depth and model's expressive power for graph heterophily/homophily. Moreover, we discover a new way to tune those weights: ReDGCN finds the proper set of weights only by adjusting its depth automatically.
> >
> > **Question 3**: Equation (7) is actually lazy random walk affinity matrix and see [6] for the generalized lazy random walk matrix.
> >
> > **Answer**: Yes. The reviewer is right. Our goal is to find a proper function $g(\cdot)$ that transforms the eigenvalues into the required range. One straightforward choice that fulfills the requirements is Equation (7), which happens to be equivalent to lazy random walk [5] [6]. From a random walk point of view, the nonzero probability of staying at the starting position (realized via self-loops) naturally contributes to the needed eigenvalue range shift. This particular form of $g(\cdot)$ serves our purpose, and we leave the designs of alternative transformation functions in future works. Also, we have added the explanations about the relation with lazy random walk and added references [5] and [6] in the revised main text.
> >
> > **Question 4**: RedGCN is essentially a linear model without any non-linearity?
> >
> > **Answer**:  No, since the diffusion process through all $d$ layers is conducted via a single operation in ReDGCN, the nonlinear function is applied after this operation in Equations (9) and (11).

---

> > > ### Author Response · Authors · 2022-11-15
> > > **Response to Reviewer NDJu (Continue)**
> > >
> > > **Question 5**: The reported results of FAGCN, GPRGNN and H2GCN in table 2 are not the same as the results in the original papers and not consistent with my personal experience.
> > >
> > > **Answer**: We would like to provide the following clarification. The performance disparity is due to the fact that we use *semi-supervised* node classification task to evaluate different models. We clarify in Section 4.1 *Experiment Setup* that for Cora, Citeseer, and Pubmed, the standard split from the original literature of GCN [1] is adopted,  and the training data ratios for these three datasets are 5.2\%, 3.6\%, and 0.3\%, respectively. For the remaining 8 datasets, under *semi-supervised* setting, we randomly split every dataset into 20\%/20\%/60\% for training/validation/testing.  This is different from the *fully-supervised* node classification setting adopted in the literature of the baselines, which split training/validation/testing into 48\%/32\%/20\% or 60\%/20\%/20\%.
> > >
> > > To further address the review's concern, in our response to Question 6, we report the performance of our ReDGCN on the standard 48\%/32\%/20\% split and compare it with the additional latest baselines mentioned by the reviewer.
> > >
> > > [1] Thomas N Kipf and Max Welling. Semi-supervised classification with graph convolutional networks. arXiv preprint arXiv:1609.02907, 2016.

---

> > > > ### Author Response · Authors · 2022-11-15
> > > > **Response to Reviewer NDJu (Continue)**
> > > >
> > > > **Question 6**: Some missing comparisons, e.g. ACM-GCN [1], LINKX [2], BernNet [3] and GloGNN [4].
> > > >
> > > > [1] Luan S, Hua C, Lu Q, et al. Is Heterophily A Real Nightmare For Graph Neural Networks To Do Node Classification?[J]. arXiv preprint arXiv:2109.05641, 2021.
> > > >
> > > > [2] Lim D, Hohne F, Li X, et al. Large scale learning on non-homophilous graphs: New benchmarks and strong simple methods[J]. Advances in Neural Information Processing Systems, 2021, 34: 20887-20902
> > > >
> > > > [3] He M, Wei Z, Xu H. Bernnet: Learning arbitrary graph spectral filters via bernstein approximation[J]. Advances in Neural Information Processing Systems, 2021, 34: 14239-14251.
> > > >
> > > > [4] Li X, Zhu R, Cheng Y, et al. Finding Global Homophily in Graph Neural Networks When Meeting Heterophily[J]. arXiv preprint arXiv:2205.07308, 2022.
> > > >
> > > > [5] Luan S, Zhao M, Hua C, et al. Complete the missing half: Augmenting aggregation filtering with diversification for graph convolutional networks[J]. arXiv preprint arXiv:2008.08844, 2020.
> > > >
> > > > [6] https://arxiv.org/pdf/2008.08844v1.pdf
> > > >
> > > > **Answer**: Thanks for the reminder of the additional latest baselines. We compare ReDGCN with ACM-GCN, LINKX, BernNet and GloGNN with the standard split of 48\%/32\%/20\%. We directly use the performances of ACM-GCN, LINKX and GloGNN reported in the ACM-GCN paper, which has just been accepted by NeurIPS'2022. For BernNet, we obtain its performance by running its open-sourced code. The results are shown in Table 1. It is observed that the latest methods, such as ACM-GCN (NeurIPS'2022) and GloGNN (ICML'2022) do perform strongly. However, our model achieves the best classfication accuracy on 7 out of the 9 datasets, which demonstrates the effectiveness of ReDGCN.
> > > >
> > > > Performance comparison (mean $\pm$ std accuracy) on benchmark datasets.
> > > >
> > > > | Datasets | Texas | Cornell | Wisconsin | Actor |Squirrel | Chameleon |
> > > > | ---- | ---- | ---- | ---- | ---- | ---- | ---- |
> > > > | BernNet | $82.70 \pm 2.7$ | $81.35 \pm 4.05$| $87.05 \pm 2.54$ | $34.46 \pm 0.77$ | $34.02 \pm 1.14$ | $47.01 \pm 1.60$  |
> > > > | LINKX | $74.60 \pm 8.37$ | $77.84 \pm 5.81$ | $75.49 \pm 5.72$ | $36.10 \pm 1.55$ | $\underline{61.81\pm 1.80}$ | $68.42 \pm 1.38$  |
> > > > |GloGNN |  $84.32 \pm 4.15$ | $83.51 \pm 4.26$ | $87.06 \pm 3.53$ | $\underline{37.35 \pm 1.30}$ | $57.54 \pm 1.39$ | $\mathbf{69.78} \pm \mathbf{2.42}$ |
> > > > |ACM-GCN |  $87.84 \pm 4.40$ | $85.14 \pm 6.07$ | $88.43 \pm 3.22$ | $36.63\pm 0.84$ | $55.19\pm 1.49$ | $\underline{69.14\pm 1.91}$ |
> > > > |RedGCN-S |  $\underline{90.32 \pm 4.71}$ | $\underline{85.81 \pm 4.21}$ | $\underline{88.63 \pm 4.51}$ | $\mathbf{37.40} \pm \mathbf{1.11}$ | $48.43 \pm 1.42$ | $67.02\pm 1.91$ |
> > > > |RedGCN-D |  $\mathbf{91.41} \pm \mathbf{3.62}$ | $\mathbf{86.53} \pm \mathbf{3.80}$ | $\mathbf{91.42} \pm \mathbf{4.22}$ | $29.03\pm 1.03$| $\mathbf{62.02} \pm \mathbf{3.03}$ | $67.33\pm 1.71$|
> > > >
> > > > | Datasets | Cora | Pubmed | Citeseer |
> > > > | ---- | ---- | ---- | ---- |
> > > > | BernNet| $87.51 \pm 0.63$ | $85.35 \pm 0.57$|$76.82 \pm 0.94$|
> > > > | LINKX  | $84.64 \pm 1.13$ | $87.86 \pm 0.77$|$73.19 \pm 0.99$|
> > > > |GloGNN |$\underline{88.31\pm 1.13}$ | $\underline{89.62 \pm 0.35}$|$\underline{77.41 \pm 1.65}$|
> > > > |ACM-GCN | $87.91\pm 0.95$ | $\mathbf{90.00} \pm \mathbf{0.52}$|$77.32 \pm 1.70$|
> > > > |RedGCN-S | $\mathbf{88.41} \pm \mathbf{1.73}$ | $85.34 \pm 2.11$|$77.42 \pm 1.30$|
> > > > |RedGCN-D | $87.90 \pm 1.31$ | $84.84 \pm 1.21$|$\mathbf{77.81} \pm \mathbf{1.72}$|
> > > >
> > > > **To better support and explain our claims, we have added all literature listed by the reviewer into the revised version of our paper. Through these listed references, we find additional works [1] [2] along the similar line of research. They are also included into our paper. Please Check the references of the revised version of our paper.**
> > > >
> > > > [1] When Do We Need GNN for Node Classification? https://arxiv.org/pdf/2210.16979.pdf.
> > > >
> > > > [2] Break the Ceiling: Stronger Multi-scale Deep Graph Convolutional Networks. NeurIPS'2019.

---

> ### Comment · Reviewer_NDJu · 2022-11-21
> **Response to Authors**
>
> After going through the response from the authors and the comments from other reviewers, I don't find a strong reason to reject this paper. Thus, I'll raise my score. Good luck.

---

> > ### Author Response · Authors · 2022-11-21
> > **Thanks to Reviewer NDJu**
> >
> > Thank you again for your constructive suggestions, agreeing with our responses, replying to our rebuttals and raising your score!!!

---

### Official Review · Reviewer_gHVm · 2022-10-25

**Confidence:** 3
**Correctness:** 4
**Technical Novelty And Significance:** 2
**Empirical Novelty And Significance:** 2
**Recommendation:** 5

**Clarity, Quality, Novelty And Reproducibility:**

This paper is well-written and well-organized. It is easy to follow the proposed ideas and technological details.
The novelty of this paper is incremental because spectrum decomposition is not new for graph analysis.
Although the proposed model achieved better performance than baselines in experiments, it is not clear what and how new information/patterns are captured by the model.

**Strength And Weaknesses:**

Strengths,

S1: This paper models the depth of GCN as a trainable parameter by eigen decomposition.

S2: Experiments validate the effectiveness of the trainable depth, especially negative depth for disassortative graphs.

Weaknesses,

W1: Adaptive and continuous depth is not new for GCN, e.g., in Graph Neural Diffusion (GRAND).

W2: Negative depth may not be a correct choice, which loses most of the topology information.

W3: This paper does not analyze the expressive power of the proposed model.



**Summary Of The Paper:**

This paper proposes a novel graph convolutional network to address the over-smoothing and heterophily issue. The proposed model is equipped with a trainable depth parameter $d$ and a selection of the spectrum of the graph signal. Based on such modeling, this paper discovers a connection between negative GCN depth and graph heterophily. The experimental results on 11 graph datasets show the proposed model outperforms baseline methods.

**Summary Of The Review:**

I recommend a weak reject for this paper because of its limited novelty and the lack of analysis of expressive power.

---

> ### Author Response · Authors · 2022-11-14
> **Response to Reviewer gHVm**
>
> **Weakness 1**: Adaptive and continuous depth is not new for GCN, e.g., in Graph Neural Diffusion (GRAND).
>
> **Answer**:
> Thanks for reminding us about the work of GRAND. However, we would like to point out there are important and subtle difference as follows. For Neural ODE and graph diffusion based works, such as CGNN [1], DGC [2], and GRAND [3], the continuous-yet-fixed integration time $T$ (equivalent to depth), even with adaptive step size [3], is not directly trainable and it cannot automatically go negative to capture heterophily, as explained in paragraph *Differences with ODE-based GNNs* in Section 3 (we add GRAND into this paragraph in the revised version). The fully trainable depth and novel functionality of negative depth has not been explored by existing works. These capabilities go far beyond making the integration time (depth) continuous with adaptive step size.
>
> Specifically, there exist 3 fundamental differences between ReDGCN and GRAND:
>
> 1. Non-trainable, non-adaptive, and predefined depth. For GRAND, the overall integration time $T$ of the ODE/PDE's solution $\mathbf{X}(T)$ serves as the depth of ReDGCN, but it is a pre-set hyper parameter, similar to $T$ in DGC and CGNN, rather than a trainable parameter. Although GRAND introduces adaptive step size $\tau$ via non-Euler multi-step schemes, such as Runge-Kutta, $T$'s value still cannot be freely changed during training.
>
> 2. Expressive power for heteropghily. The non-trainable positive integration time/depth $T$ in GRAND doesn't automatically go negative when graph heterophily exists. Thus, GRAND doesn't have the expressive power of processing high-frequency graph signals, while ReDGCN can.
>
> 3. Model complexity. The performance of GRAND on homophilic graphs (no results on heterophilic graphs are presented in the paper) largely depends on the multi-head attention mechanism. Each of the multiple scaled-product attention heads needs a set of parameter matrices $\mathbf{W}_Q$ and $\mathbf{W}_K$, which significantly increases the number of parameters and model complexity. In contrast, the number of parameters in ReDGCN equals to that of a 1-layer vanilla GCN plus 1 (ReDGCN-S) or 2 (ReDGCN-D) depth parameters.
>
> [1] Louis-Pascal Xhonneux, Meng Qu, and Jian Tang. Continuous graph neural networks. In International Conference on Machine Learning, pp. 10432–10441. PMLR, 2020.
>
> [2] Yifei Wang, Yisen Wang, Jiansheng Yang, and Zhouchen Lin. Dissecting the diffusion process in linear graph convolutional networks. Advances in Neural Information Processing Systems, 34: 5758–5769, 2021.
>
> [3] Ben Chamberlain, James Rowbottom, Maria I Gorinova, Michael Bronstein, Stefan Webb, and Emanuele Rossi. Grand: Graph neural diffusion. In International Conference on Machine Learning, pp. 1407–1418. PMLR, 2021.

---

> > ### Author Response · Authors · 2022-11-14
> > **Response to Reviewer gHVm (Continue)**
> >
> > **Weakness 2**: Negative depth may not be a correct choice, which loses most of the topology information.
> >
> > **Answer**: We are afraid that the reviewer mis-understood this and would like to provide the following clarifications.
> >
> > The negative depth and how it exploits the topological information can be explained in the following aspects:
> >
> > 1. Graph spectrum theory. The effectiveness of negative depth on processing high-frequency graph signals for heterophilic graphs is proven by rigorous mathematics in Section 3.
> >
> > 2. Message propagation. In the *Remark* paragraph of the Model section (renamed as *Negative depth explained* in the revised paper), we explain that a positive depth propagates message forward, while a negative depth cancels the effect of the forward diffusion; In case of $d=-1$, it physically constructs a different diffusion matrix ($\mathbf{S}^{-1}$) and conducts one-step message diffusion. The topological information is fully utilized both in forward and backward propagation processes.
> >
> > 3. Graph topology. In Section 4.4, the spectral and geometric properties brought by the negative depth are analyzed in detail, so that the physical effects generated by negative depth are further clarified: it strengthens the self-loops (the ``staying-on-node'' probabilities from the random walk point of view) and reduces the impact of large degree nodes.  In addition, as the depth becomes negative, it means mainly keeping the **eigengraphs corresponding to high frequency** (See Figure 2(b) and Figure(5)), which have more **negative** weighted edges. So, it is equivalent to turning some edges with positive weights in the original graph to modified edges with negative weights in the augmented graph, which can make node  embeddings  dissimilar in the message-passing process and naturally handle the heterophilic graphs (See Figure 1). Actually, RedGCN **finds a better way to utilize the graph topology to tackle the heterophilic graph**.

---

> > > ### Author Response · Authors · 2022-11-14
> > > **Response to Reviewer gHVm (Continue)**
> > >
> > > **Weakness 3**: This paper does not analyze the expressive power of the proposed model.
> > >
> > > **Answer**:
> > > Our methodology is motivated and designed based on the analysis of spectral GNNs' expressive power, as presented in the first half of Section 3. As well-studied in previous works [1] [2] [3], the expressive power of GNNs is equivalent to the ability of extracting and filtering low or high frequency components that characterizing the input graph signals. It also reflects the model's capability of expressing graph homophily and heterophily. Due to the low-pass nature of many GNN models, such as GCN/SGC, their expressive power is for modeling homophilic graphs, which mainly contain low-frequency signals. Other models extend their expressive power to heterophilic graphs via sophisticated structures (high-pass) or two-channel architectures (low- and high-pass) [2] [3]. However, our model obtains comparable or higher expressive power by automatically tuning the depth.
> > >
> > > Moreover, the expressive power of GNNs can also be represented by their frequency profiles [1] [2]. While the monoticities of most existing models' frequency profiles are hard to change, the adjustable depth in our model provides us the freedom to tune the frequency profile to best match the underlying graph signal, as shown in Figures 2 and 5.
> > >
> > > To better clarify the expressive power of our model, we have made corresponding changes in Section 3 of the revised paper to explicitly specify the relation between its expressive power and its frequency filtering properties.
> > >
> > > [1] Muhammet Balcilar, Guillaume Renton, Pierre He ́roux, Benoit Gau ̈ze`re, Se ́bastien Adam, Paul Honein.Analyzing the Expressive Power of Graph Neural Networks in a Spectral Perspective. ICLR 2021.
> > >
> > > [2] Deyu Bo, Xiao Wang, Chuan Shi, and Huawei Shen. Beyond low-frequency information in graph convolutional networks. In Proceedings of the AAAI Conference on Artificial Intelligence.
> > >
> > > [3] Sitao Luan, Mingde Zhao, Chenqing Hua, Xiao-Wen Chang, Doina Precup. Complete the missing half: Augmenting aggregation filtering with diversification for graph convolutional networks. NeurIPS 2022 GLFrontiers Workshop.

---

> > > > ### Author Response · Authors · 2022-11-14
> > > > **Response to Reviewer gHVm (Continue)**
> > > >
> > > > **Weakness in Clarity, Quality, And Reproducibility**
> > > >
> > > > **Weakness 1**: The novelty of this paper is incremental because spectrum decomposition is not new for graph analysis.
> > > >
> > > > **Answer**: Spectrum decomposition serves as a common tool set for many spectral GNN works, including our work. We use this tool set in a novel way to redefine depth and to analyze our model's frequency profile. The novelty of our work lies in the new understanding of GCN's depth, which is adjustable and can automatically go negative for heterophilic graphs. To the best of our knowledge, no previous methodology applies spectrum decomposition to manipulate GNN's depth and to explicitly model graph heterophily via negative depth.
> > > >
> > > > **Weakness 2**: Although the proposed model achieved better performance than baselines in experiments, it is not clear what and how new information/patterns are captured by the model.
> > > >
> > > > **Answer**: As explained in the response to Question 3, via adjustable depth, our model finds (1) the suitable frequency profile and (2) the required expressive power matching the underlying graph signal. By this means, as a graph signal filter, the essential frequency components (the key ``information/pattern'') of the input graph signals are well captured, while the non-existing ones are suppressed.
> > > >
> > > > Moreover, the effectiveness of our method can be understood from a graph augmentation point of view. As explained in Section 4.4, the augmented graph obtained via the learned depth has a novel topology. Such a topology strengthens the self-loops and reduces the impact of large degree nodes, so that the message propagation between heterophilic nodes are suppressed, which helps keep the ability to be distinguished among the dissimilar nodes.

---

> ### Author Response · Authors · 2022-11-23
> **Looking forward to discussing**
>
> Dear Reviewer gHVm, since the deadline for Reviewer-Author discussion ended five days ago, we will be very grateful if you could take a look at our responses to your comments, including the differences with GRAND, the expressiveness of our method, the clarification about the meaning of the negative depth and the learned patterns of ReDGCN, e.t.c. Looking forward to discussing with you~ Thanks.

---

### Official Review · Reviewer_ZTa3 · 2022-10-25

**Confidence:** 3
**Correctness:** 3
**Technical Novelty And Significance:** 3
**Empirical Novelty And Significance:** 2
**Recommendation:** 5

**Clarity, Quality, Novelty And Reproducibility:**

The problem is interesting and the proposed method is novel. But the physical meaning of negative results is not clear.

**Strength And Weaknesses:**

Strength: The problem is interesting and usefull for GCN designing.
Weaknesses: The motivation and the physical meaning of the negative depth value are not clear. When the depth of GCN is negative, what dose it mean?

**Summary Of The Paper:**

In this work, by redefining GCN’s depth d as a trainable parameter continuously adjustable within positive infinity and negative infinity, a simple and powerful GCN model RED-GCN is proposed to retain the simplicity of GCN and meanwhile automatically search for the optimal d without the prior knowledge regarding whether the input graph is homophilic or heterophilic.

**Summary Of The Review:**

The problem is interesting and the proposed method is novel. But the physical meaning of negative results is not clear.

---

> ### Author Response · Authors · 2022-11-14
> **Response to Reviewer ZTa3**
>
> **Question 1**: The motivation and the physical meaning of the negative depth value are not clear. When the depth of GCN is negative, what does it mean?
>
> **Answer**:
>
> The intuitive explanations of the physical meaning of the negative depth have been presented in both Model and Experiment sections in the paper. We focus mainly on two aspects:
>
> (1) Message propagation: in the *Remark* paragraph of the Model section, we explain that a positive depth propagates message forward, while a negative depth cancels the effect of the forward diffusion. In case of $d=-1$, it physically constructs a different diffusion matrix ($\mathbf{S}^{-1}$) and conducts one-step message diffusion.
>
> (2) Graph topology: in Section 4.4, the spectral and geometric properties brought by the negative depth are analyzed in detail, so that the physical effects generated by negative depth are further clarified: it strengthens the self-loops (the ``staying-on-node'' probabilities from the random walk point of view) and reduces the impact of large degree nodes.  In addition, as the depth becomes negative, it means mainly keeping the **eigengraphs corresponding to high frequency** (See Figure 2(b) and Figure(5)), which have more **negative** weighted edge. So, it is equivalent to turning some edges with positive edge weights in the original graph to modified edges with negative weights in the augmented graph, which can make node  embeddings  dissimilar in the message-passing process and naturally handle the heterophilic graphs (See Figure 1).
>
> To help the readers locate those explanations more easily, in the Model section, we have changed the corresponding paragraph title from *Remark* to *Negative depth explained* in the revised main text. This paragraph also points to Section 4.4 for further insights on the related geometric properties.

---

> ### Author Response · Authors · 2022-11-23
> **Looking forward to discussing**
>
> Dear Reviewer ZTa3, since the deadline for Reviewer-Author discussion has ended five days and you only have one question about the meaning of the negative depth, we will be very grateful if you could have a look at the clarification of the meaning of negative depth we made here and in the main contents of the paper, which we believe can help you better understand it. Thank you very much~

---

### Author Response · Authors · 2022-11-14
**Overall Response.**

First, we thank all the reviewers for your time and efforts spent on reviewing our paper and the given constructive comments and suggestions! Hope our responses can address your concerns. We also have submitted a revised version of the paper, including the following changes, which are marked with blue color:

(1) Add some clarification sentences in the main text regarding to explanation of negative depth, expressive power and differences with GRAND paper.

(2) Add all the papers listed by reviewers and some other related works as references.

(3) Add a visualization of node embedding generated by RedGCN-S under different depths for homophilic (Cora) and heterophilic (Texas) graphs and some related insights/analysis in Appendix. Please download the supplementary file to check.

---

### Author Response · Authors · 2022-11-20
**Looking forward to discussing with all reviewers**

We sincerely thank the reviewers for providing valuable comments. Since the deadline (11/18) for rebuttal and discussion has already ended but we haven't received any comments/feedbacks on our rebuttal/responses from any reviewers. Hopefully,  if you had a chance to go over our responses, please let us know if we have addressed all your concerns and questions or not. We are looking forward to a fruitful discussion and your replies, which are highly valuable to us. We would appreciate it if you consider increasing your scores if our response can address your questions.

---

### Decision · Program_Chairs · 2023-01-20

**Decision:**

Reject

**Justification For Why Not Higher Score:**

The reviewers suggested during the meeting that this paper could benefit in using synthetic experiments to substantiate their claims.

**Justification For Why Not Lower Score:**

N/A

**Metareview: Summary, Strengths And Weaknesses:**

__Summary.__ The authors generalize the depth of the GCN to a continuous (real) value d, which is trainable and adjustable. To justify this approach, the authors show through an eigen decomposition of the symmetrically normalized graph Laplacian, a correlation between graph homophily/heterophily and the eigenvector frequencies. This analysis allows them to introduce the notion of eigengraph – a weighted graph with potentially negative edges: the normalized adjacency matrix can in fact be written as a linear combination of such eigengraphs, where the coefficients determine the ability of GCN to capture homophilic/heterophilic signals. This eigengraph analysis of the convolution leads the authors to consider its extension to arbitrary depths. The authors then show through a set of experiments that heterophilic graphs are usually associated with low (or even negative) diffusion depths.

__Summary of the reviews.__  Overall, the reviewers’ feedback for this submission was tepid at best. The contribution does not seem extremely novel (continuous depth was already introduced by GRAND, but the possible negativity of the depth is new to this submission). One of the issues brought forward by the reviewers was the difficulty in scaling up these methods, solved to some extent in the paper by arbitrarily selected the K lowest and K highest eigenvalues. The authors’ experiments lack in showing the applicability of their method to large scale graphs. The main issue with the evaluation of the method is that it is performed on real life graphs, which are necessarily noisy and imperfect. To really verify their claims, the reviewers would have preferred the authors to evaluate their methods on synthetic datasets (eg Stochastic Block Models), where the degree of homophily and heterophily can be controlled much more accurately. __For this reason, we recommend a rejection.__

**Summary Of Ac-Reviewer Meeting:**

The reviewers expressed their tepid enthusiasm for the paper: the notion of continuous depth is not novel ---but that of negative is. While the latter is potentially interesting, one reviewer recommended a stronger suit of experiments to evaluate this approach. In particular, the authors could use Stochastic Block Models to generate homophilic and heterophilic networks, and evaluate their method in a much more controlled environment. As such, the point that the authors are trying to make is not well enough substantiated by experiments, and the reviewers generally considered this paper to be extremely borderline as such.